# A nationwide epidemiological study of testicular torsion: Analysis of the Japanese National Database

Aya Hiramatsu[1,2], Hiroki Den[1]*, Masashi Morita[2], Yoshio Ogawa[3], Takashi Fukagai[3], Akatsuki Kokaze[1]

1 Department of Hygiene, Public Health, and Preventative Medicine Showa University School of Medicine, Shinagawa-ku, Tokyo, Japan, Japan, 2 Department of Urology Showa University Koto Toyosu Hospital, Koto-ku, Tokyo, Japan, 3 Department of Urology Showa University School of Medicine, Shinagawa-ku, Tokyo, Japan

* denh@med.showa-u.ac.jp

## Abstract

Testicular torsion is a severe urological emergency caused by the twisting of the spermatic cord. The nationwide incidence of testicular torsion in Japan has not been previously reported. Accordingly, we aimed to estimate the nationwide incidence of testicular torsion using the National Database of Health Insurance Claims and Specific Health Checkups of Japan (NDB) and examine the orchiectomy rate. This cross-sectional study was based on data from the NDB. We extracted data of patients aged < 21 years with documented testicular torsion and relevant treatment from January 2018 to December 2020. Testicular torsion was identified based on the Japanese standardized disease codes. The national incidence rate between 2018 and 2020 was calculated and assessed according to age and region of origin. Orchiectomy rates were evaluated according to age. The nationwide incidence rates of testicular torsion were 14.46, 15.09, and 15.88 per 100,000 males aged < 21 years in 2018, 2019, and 2020, respectively. The orchiectomy rate was 7.1%. Testicular torsion was most frequently observed during winter. A similar trend was observed nationwide. To the best of our knowledge, this study is the first to report the nationwide incidence of testicular torsion in Japan.

## Introduction

Testicular torsion (TT) is a severe urological emergency caused by the twisting of the spermatic cord, resulting in interrupted blood supply and testicular ischemia. Delayed care may necessitate an orchiectomy, which can lead to decreased testicular function [1].

TT occurs when the testis rotates around its spermatic cord attachments, preventing blood flow and resulting in tissue ischemia [2]. With prolonged ischemia, the involved testes can experience irreversible injury, atrophy, and loss of function. When TT is diagnosed, or strongly suspected, urgent scrotal exploration and detorsion of the spermatic cord are indicated [2]. If the torsion is released and blood reperfusion is confirmed, "spermatic cord torsion

MHLW. Access to the raw data is restricted, requiring prior approval from the MHLW. According to the guidelines, all data must be deleted after the research is completed. For further information, please contact: Division for Health Care and Long-term Care Integration, Health Insurance Bureau, Ministry of Health, Labour and Welfare, Japan Tel: +81-3-5253-1111.

**Funding:** The author(s) received no specific funding for this work.

**Competing interests:** The authors have declared that no competing interests exist.

surgery" is performed, and if necrosis is observed, "orchiectomy" is performed. Orchiopexy, which involves testicular fixation to the inner scrotal wall, can prevent further TT. Previously, contralateral orchiopexy was frequently suggested owing to the typical presence of a bell clapper deformity in the contralateral testicle [3].

Testicular salvage rates of 90–100% have been documented when TT is addressed within 4–8 h of symptom onset; however, the rates reduced to 50 and 10% at 12 and 24 h of symptom onset, respectively [4]. Median times from symptom onset to surgery have been reported as 5 and 48 hours in the testis-sparing and orchiectomy groups, respectively, indicating a substantially prolonged time in the group undergoing testicular removal [5]. Accordingly, to improve testicular salvage rates, young males with testicular pain should be immediately hospitalized [6].

Reportedly, 26% of patients with acute scrotum are diagnosed with TT [7]. Hatano et al. reported that the annual incidence rate of TT was approximately 0.56 patients per 100,000 males in Kyusyu, the third-largest island in Japan among the five main islands and the southernmost of the four largest islands [8]; however, to the best of our knowledge, the epidemiology of TT and the risk factors for orchiectomy after TT have not been explored in a large population in Japan, with most published studies involving single-institution retrospective reviews [5,9].

Herein, we aimed to improve the current knowledge regarding the epidemiology of TT.

We examined the nationwide incidence of TT using data from one of the world's largest health-related databases, the National Database of Health Insurance Claims and Specific Health Checkups of Japan (NDB), and determined the orchiectomy rate in Japan. The aim of this study was to investigate the nationwide epidemiological and clinical features of TT.

## Materials and methods

### Data source

This was a cross-sectional study based on NDB data. Japan has implemented a compulsory nationwide public health insurance system for all legal Japanese citizens, excluding short-term residents (length of stay less than 90 days) and those on welfare. The citizens have access to covered medical services when consulting with a doctor. Medical institutions submit patient claims to the national insurance program for payment of expenses for the covered services [10,11]. Since 2009, under relevant legislation, the Japanese Ministry of Health, Labour, and Welfare (MHLW) has retained all insurance claim data, except those related to clinical trials as well as medical expenses that are not covered by insurance. The data are stored in the NDB. Since 2011, the MHLW began providing this data for research projects and to local governments for health policy decision making [10,12]. The NDB contains complete datasets of medical care provided to insured inpatients and outpatients [13]. The NDB contains data of an extremely large study population, including sufficient sample size of individuals with relatively rare conditions, such as TT.

The use of claims data of patients with TT stored in the NDB was approved by the MHLW in October 2021 (No. 1424). This study was approved by the Ethics Committee of Showa University School of Medicine (Tokyo, Japan) in August 2021 (No. 21-001-B). The data has already been fully anonymized by the MHLW before our access; therefore, the requirement of informed consent was waived for the current study. This study included individuals whose data were stored in the NDB.

The NDB dataset contains various data elements, including personal identifiers (replaced with an identification data [ID] variable), dates, age groups, gender, procedure descriptions, International Classification of Diseases 10th Revision (ICD-10) diagnosis codes from the World Health Organization, details of medical care, records of medical examinations without

reported results, and information about prescribed medications. Notably, this information was sourced independently of input from healthcare providers or patients.

To contextualize our study, since the NDB operates as a claims-based database, we acknowledge the potential for discrepancies between the recorded disease names and the actual medical conditions. To enhance the precision of the extracted data and ensure more accurate data, we considered it necessary to adopt a dual approach, considering both the disease name and the surgical procedure. When a diagnosis of TT is established, it invariably necessitates a surgical intervention. To discern and exclude spurious diagnoses, we contemplated the extraction of IDs encompassing both the disease denomination and the associated surgical procedure.

We received the dataset from the MHLW on March 7, 2023, and extracted data of patients aged < 21 years with documented TT and relevant treatment from January 2018 to December 2020. TT was identified based on the Japanese standardized disease codes. TT and other TT-related disease names, codes, and their corresponding ICD-10 codes were as follows: TT (6082005/8835916, ICD-10: N44), epididymal torsion (8850247, ICD-10: N44), epididymal torsion (8835904, ICD-10: N44), epididymal torsion (8850248, ICD-10: N44), testicular pain (8835913, ICD-10: N508), and scrotal pain (8830696, ICD-10: N508).

For our research study, we opted to extract disease names beyond TT, considering that the NDB data were manually input by healthcare providers. This decision was based on the awareness that, despite best practices, there is a potential for inputting omissions, which prompted the inclusion of a wider spectrum of disease names for more comprehensive data inclusion. However, in practice, in very few instances was TT surgery performed for diseases other than TT. Such cases were not included in this study.

We extracted IDs for patients aged < 21 years using the above disease codes and the following treatment codes: spermatic cord torsion surgery (150254510), spermatic cord torsion surgery with contralateral testicular fixation (150297710), and orchiectomy (150207510). In spermatic cord torsion surgery, only the torsion testicle is fixed to the scrotal floor without contralateral testicular fixation. Medical claims assigned to the same ID1 were considered claims for the same patient. In addition, ID1 may change when a patient's health insurance card number changes. Therefore, medical claims assigned different ID1s, but the same ID2, were also considered claims for the same patient. Only the first diagnosis was included if a patient had been treated more than once during the same period. We excluded cases in which disease or medical treatment was related to trauma or a malignant tumor. The extracted IDs were categorized into three groups based on the disease names: TT only, TT along with other TT-related disease names, and other TT-related disease names only. Patients who underwent surgery for other TT-related disease names were excluded from the analysis (Fig 1).

The NDB guidelines stipulate that when the number of cases is < 10, estimates cannot be reported, to protect the individuals involved. Accordingly, Japanese prefectures were divided into seven regions based on an established convention [12], and estimates were reported for these regions. Japan is a geographically elongated country, extending from the north to south. Hokkaido/Tohoku (Region 1) is located at a higher latitude, experiencing colder weather, while Kyusyu/Okinawa (Region 7), situated at a lower latitude, tends to have a warmer climate. As a result, there are significant climatic variations across the country. First, the national incidence rate between 2018 and 2020 was calculated. To calculate the incidence rate, the nationwide and prefectural population data were obtained from e-Stat, the Japanese government statistics site [14]. Second, the incidence rates were assessed according to the age and region of origin. Orchiectomy rates were evaluated according to age. The current study defined March-May, June-August, September-November, and December-February as spring, summer, fall, and winter, respectively, based on the Japan Meteorological Agency's definitions [15]. Given

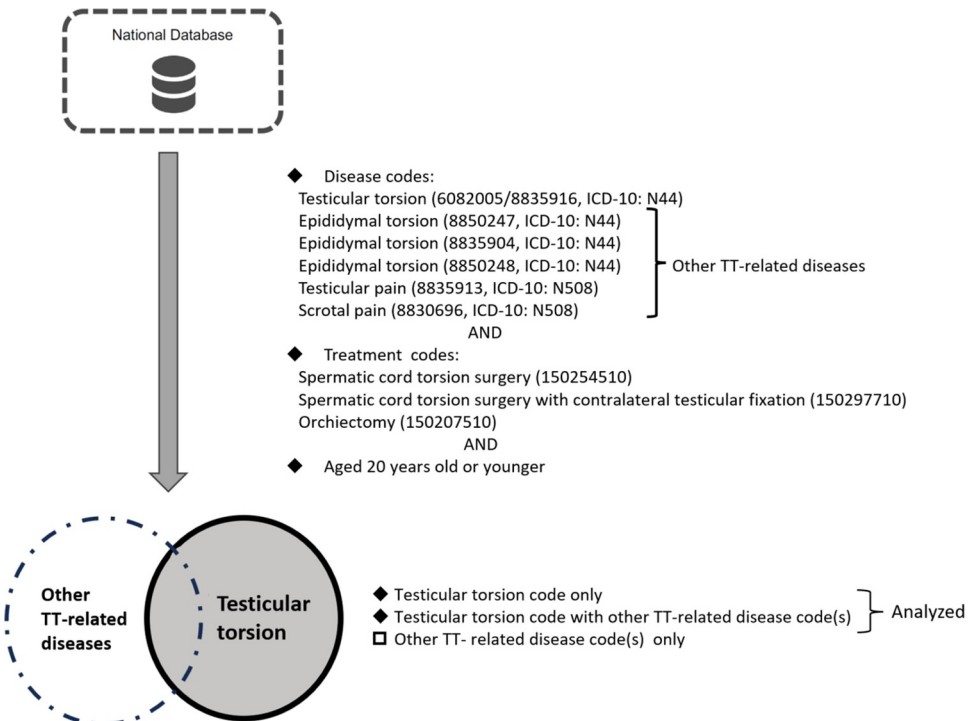

**Fig 1. Data extraction methods.** From the database, records meeting the following conditions were extracted: a) containing TT or other TT-related disease codes, b) including treatment codes for TT, and c) aged 20 years old or younger. Data indicating other TT-related disease codes only were excluded from the analysis.

that climate varies during each season, the effect of the season on the incidence of TT was also evaluated.

## Statistical analysis

All statistical analyses were performed using the R software version 4.2.2 (R Foundation for Statistical Computing, Vienna, Austria). The chi-squared test was used to compare categorical data. When the chi-squared test statistic was significant, a pairwise chi-squared test with a Bonferroni correction was conducted. The odds ratio was estimated to assess differences in the incidence rate of TT between summer and winter. A p-value <0.05 was considered statistically significant.

## Results

In total, 1668, 1714, and 1779 patients were diagnosed with TT in 2018, 2019, and 2020, respectively. Numbers of other TT-related diseases are in Table 1. While cases of co-occurrence of TT and undescended testes were observed, they were limited to fewer than 10 cases per year. The nationwide incidence rate of TT was 15.13 per 100,000 males aged < 21 years between 2018 and 2020 (Table 2). Overall, 1368, 2756, 140, and 897 patients had right, left, bilateral, and unknown regions of origin TTs, respectively. The left side was more frequently affected than the right side (2:1 ratio).

The incidence rate of TT per 100,000 individuals was significantly higher in the 10–14 years age group than that in any other age group (p < 0.001) (Fig 2).

**Table 1. Numbers of diseases.**

| Disease names (disease name codes) | 2018 | 2019 | 2020 |
|---|---|---|---|
| Epididymal torsion (8835904, ICD-10: N44) | 15 | 18 | <10 |
| Epididymal torsion (8850247, ICD-10: N44) * | 0 | 0 | 12 |
| Epididymal torsion (8850248, ICD-10: N44) * | 0 | 0 | 0 |
| Testicular pain (8835913, ICD-10: N508) | 0 | 0 | <10 |
| Scrotal pain (8830696, ICD-10: N508) | 0 | 0 | 0 |
| Testicular torsion (6082005/8835916, ICD-10: N44) | 1668 | 1714 | 1779 |

*These disease name codes were newly established in the year 2020.
The numbers for other TT-related diseases represent cases where treatment for TT was also administered. The guidelines indicate that cases with fewer than 10 occurrences cannot be disclosed.

According to the data, 24.5% of patients underwent spermatic cord torsion surgery, 68.4% underwent spermatic cord torsion surgery with contralateral testicular fixation, 0.9% underwent spermatic cord torsion surgery with contralateral testicular fixation and orchiectomy, and 6.2% underwent orchiectomy.

The overall orchiectomy rate between 2018 and 2020 was 7.1%, which was significantly higher in the 0–4 age group than in any other groups ($p < 0.001$) (Fig 2). TT occurred most frequently during winter, with a similar trend observed nationwide. The number of occurrences significantly differed between summer (June-August) and winter (December-February). Winter had a more significant effect on the incidence of TT than summer (odds ratio, 1.42; 95% confidence interval, 1.24–1.63) (Fig 3). Region 2 had a significantly higher TT incidence rate than Regions 3, 6, and 7 ($p < 0.001$, 0.03, and <0.001, respectively). Furthermore, statistically significant differences were only observed in Regions 2 and 3, although the highest incidence was documented nationwide during winter (Table 3).

## Discussion

To the best of our knowledge, this is the first study to report the nationwide incidence rate of TT in Japan, which was 15.13 per 100,000 males aged < 21 years in 2018–2020.

We extracted data of males aged < 21 years because incidence of TT reportedly peaks during infancy and adolescence, and occurrence in older age groups is substantially rare [4,16,17]. Moreover, enhancing testicular preservation among this generation will help maintain fertility rates.

The reported nationwide incidence rates in various countries are as follows. The incidence rates of TT in the USA was 3.8 per 100,000 male participants aged < 18 years in 2006 [16]; approximately 1.4 per 100,000 males in 2010 in Brazil [18]; 21.76 cases per 100,000 person-years aged < 25 from 2009 to 2018 in Ireland [19]; 2.9 per 100,000 person-years in males

**Table 2. Incidence of testicular torsion.**

| Calendar year | No. of TT | Male individuals aged < 21 yrs. (thousands) | Incidence rate (per 100,000 person-years) |
|---|---|---|---|
| 2018 | 1668 | 11,537 | 14.46 |
| 2019 | 1714 | 11,362 | 15.09 |
| 2020 | 1779 | 11,202 | 15.88 |
| Total | 5161 | 34,101 | 15.13 |

No., number; TT, testicular torsion; yrs., years.

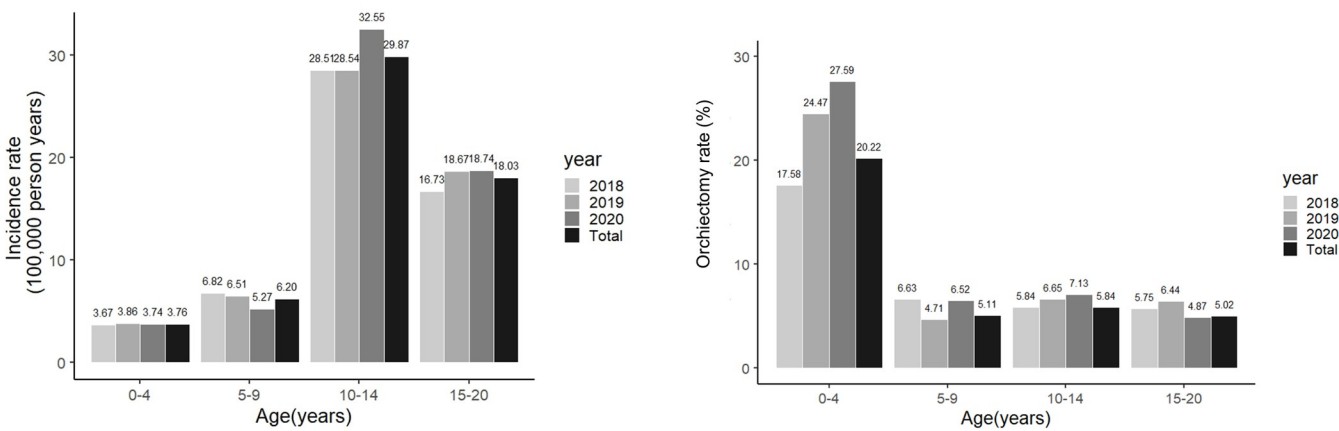

**Fig 2.** Incidence rate of testicular torsion according to age distribution (Left). Orchiectomy rate according to age distribution (Right).

aged < 25 years from 2006 to 2011 and 6.99 per 100,000 person-years in those aged < 19 years from 2009 to 2019 in Korea [17,20]; and 3.5 per 100,000 person-years in Taiwan [21]. Although undertaking a direct comparison can be challenging owing to distinct population compositions, the current study revealed that the incidence rate of TT in Japan is relatively higher than that previously reported in other countries. The observed discrepancy could be attributed to the small financial burden of universal health insurance that may have facilitated

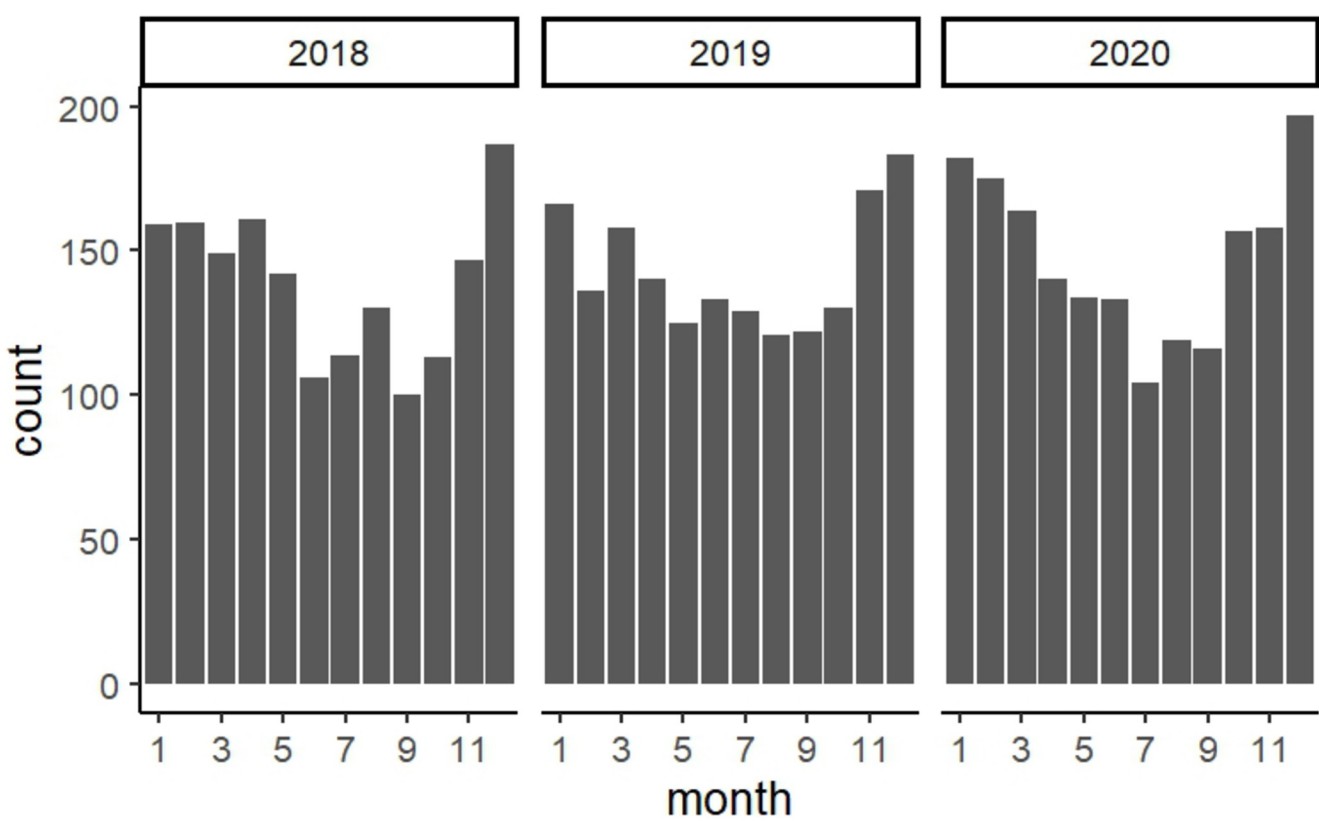

**Fig 3. Incidence of cases per month.**

**Table 3. Incidence rate of TT according to region in Japan (per 100,000 annually, <21 years).**

| Region | 2018 | 2019 | 2020 | Total | OR (95%CI) (winter effect) |
|---|---|---|---|---|---|
| 1.Hokkaido/Tohoku | 14.18 | 14.70 | 15.59 | 14.69 | 1.36 (0.87–2.15) |
| 2.Kanto | 16.50 | 17.03 | 17.18 | 16.85 | 1.49 (1.19–1.88) * |
| 3.Chubu | 13.28 | 13.88 | 14.39 | 13.76 | 1.55 (1.10–2.21) * |
| 4.Kinki | 14.15 | 14.44 | 17.87 | 15.38 | 1.37 (0.99–1.90) |
| 5.Chugoku | 12.43 | 16.72 | 14.74 | 14.52 | 1.65 (0.92–3.03) |
| 6.Shikoku | 14.77 | 10.15 | 12.81 | 12.45 | 1.00 (0.35–2.84) |
| 7.Kyusyu/Okinawa | 14.03 | 13.18 | 13.14 | 13.42 | 1.13 (0.74–1.74) |
| Total | 14.46 | 15.09 | 15.88 | 15.13 | 1.42 (1.24–1.63) * |

Region 1 (Hokkaido and Tohoku: Hokkaido, Aomori, Iwate, Miyagi, Akita, Yamagata, and Fukushima), Region 2 (Kanto: Ibaraki, Tochigi, Gunma, Saitama, Chiba, Tokyo, and Kanagawa), Region 3 (Chubu: Niigata, Toyama Ishikawa, Fukui, Yamanashi, Nagano, Gifu, Shizuoka, and Aichi), Region 4 (Kinki: Mie, Shiga, Kyoto, Osaka, Hyogo, Nara, and Wakayama), Region 5 (Chugoku: Tottori, Shimane, Okayama, Hiroshima, and Yamaguchi), Region 6 (Shikoku: Tokushima, Kagawa, Ehime, and Kochi), Region 7 (Kyushu and Okinawa: Fukuoka, Saga, Nagasaki, Kumamoto, Oita, Miyazaki, Kagoshima, and Okinawa).

TT, testicular torsion; OR, odds ratio; CI, confidence interval.

Incidence rates were compared according to region. The incidence is higher in winter (December to February) than in summer (June to August).

*Significant difference.

better access to hospital visits. Additionally, some local governments subsidize medical expenses for minors, i.e., the age at which TT occurs most frequently, thereby establishing a system in which medical services are available regardless of income disparity.

In this study, the total testicular salvage rate was 92.9% (orchiectomy rate, 7.1%). The orchiectomy rate was particularly high in the 0–4 age group (20.22%). More recently, testicular salvage rates ranging between 58.1 and 75.7% (orchiectomy rate, 24.3–41.9%) have been reported [8,16,20,21]. In South Korea, removal rates of 45.37 and 20.09% have been documented among 0–1 and 2–9 year-old males, respectively [17]. As mentioned earlier, one of the risks associated with orchiectomy during TT is the delay in timing [2].

A possible reason for the low orchiectomy rate in Japan is the short time between onset and surgery, which can be attributed to the high accessibility of emergency room visits and specialists by patients, as well as the high surgical acceptance in hospitals. However, similar to the high incidence of TT, the low orchiectomy rate may also be due to overdiagnosis.

In the case of patients under 4 years, in particular, their high rate of orchiectomy is consistent with other reports [22]. This could be attributed to the fact that those patients often exhibit limited symptoms, leading to delayed and inaccurate diagnosis [17,23].

Nevertheless, the orchiectomy rate for infants in Japan may be lower than rates reported in other countries.

In the present study, the incidence rate of TT was higher between December and February (winter) than that observed during any other season, which is consistent with the findings of previous studies [24]. Moreover, the high TT incidence during winter was observed across all regions. Regions 2 and 3 showed significant differences, although differences were not significant in other regions owing to low incidence rates. There were significant differences in incidence rates by region. Region 2 had a significantly higher incidence than Regions 3, 6, and 7. However, although this result is statistically significant, a clinical difference of approximately 2 per 100,000 individuals is not considered significant in practice.

In 1983, Shukla et al. reported a substantial correlation between the incidence of TT and ambient temperature < 2°C in Ireland [25]. Subsequently, environmental risk factors and seasonal associations for TT were reported. In the USA, the incidence rates of TT during the

spring and winter accounted for 67.2% of cases when compared to those during fall and summer [24]. TT follows a seasonal association even in tropical countries, such as Brazil, and is reportedly more frequent during the colder months [26]. In Japan, Hoshino et al. and Takeshita et al. have reported that an ambient temperature < 15˚C at pain onset was a risk factor for developing TT [27,28].

Although the NDB data did not allow for the evaluation of the outside temperature at the time of diagnosis owing to difficulties in investigating the temperature at the time of diagnosis, the above results suggest that the outside temperature is likely to influence the incidence of TT.

This study revealed that left-sided TT had a significantly higher incidence rate than right-sided TT, with a 2:1 ratio. This is consistent with previous reports [9,29,30], given that the left spermatic cord is often slightly longer than the right, making the left testis more mobile and, consequently, more susceptible to torsion than the right testis. This anatomical difference has been associated with the left-sided preponderance of TT [30]. Bilateral TT is extremely rare [31], and it is likely that several of these cases are entered in receipt data as disease names when performing contralateral fixation.

This study has some limitations because diagnosis and treatment were analyzed based on inferences from the NDB data. Therefore, some cases may deviate from the actual diagnosis and treatment. In the following scenarios, there is a potential for a mismatch between the actual disease name and the NDB.

First, surgery might be performed upon a suspicion of TT, which is subsequently determined not to be TT. Alternatively, even when it is known not to be TT, surgery for TT might be conducted as a preventive measure. As long as the TT disease code is still included, distinguishing between actual occurrence and false diagnosis is challenging and cases might be overestimated. For example, undescended testes are more likely to be complicated by TT [32,33]. Considering prophylactic contralateral fixation during surgery for undescended testis, some cases may be registered as a disease even in the absence of TT. However, it is worth noting that fewer than 10 such cases have been reported annually, and their impact on the overall incidence may be limited. Nevertheless, it is likely that the existence of a difference between the actual number of medical conditions and the recorded diagnoses cannot be eliminated.

Furthermore, although the NDB stores > 90% of medical claims [12], some patients with TT might not have been included. Consequently, the incidence rates reported in the present study may have been underestimated.

Despite these limitations, the NDB remains a useful data source for epidemiological studies, allowing the estimation of the nationwide incidence of TT and associated risk factors for this rare disease.

In summary, the present study is the first to estimate the nationwide incidence rate of TT in Japan using the NDB. The incidence rate of TT was 15.13 per 100,000 males aged < 21 years and the orchiectomy rate was 7.1%. Our findings revealed a higher TT incidence rate than that reported previously. Owing to its low incidence, TT is a disease with unknown features. Accordingly, this study may help to clarify the pathogenesis of TT.

## Acknowledgments

This study utilized data from the National Database of Health Insurance Claims and Specific Health Checkups of Japan, which is under the jurisdiction of the MHLW. Access to the raw data is restricted, requiring prior approval from the MHLW. According to the guidelines, all data must be deleted after the research is completed.

## Author Contributions

**Conceptualization:** Aya Hiramatsu, Hiroki Den.

**Data curation:** Aya Hiramatsu, Hiroki Den.

**Formal analysis:** Hiroki Den.

**Investigation:** Aya Hiramatsu.

**Methodology:** Aya Hiramatsu, Hiroki Den.

**Project administration:** Aya Hiramatsu.

**Resources:** Aya Hiramatsu.

**Software:** Hiroki Den.

**Supervision:** Hiroki Den.

**Validation:** Hiroki Den.

**Visualization:** Hiroki Den.

**Writing – original draft:** Aya Hiramatsu.

**Writing – review & editing:** Hiroki Den, Masashi Morita, Yoshio Ogawa, Takashi Fukagai, Akatsuki Kokaze.

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
