## [Decision Letter · Decision Letter 0]

25 Oct 2023

PONE-D-23-27667A nationwide epidemiological study of testicular torsion: analysis of the Japanese National DatabasePLOS ONE

Dear Dr. Den,

Thank you for submitting your manuscript to PLOS ONE. After careful consideration, we feel that it has merit but does not fully meet PLOS ONE’s publication criteria as it currently stands. Therefore, we invite you to submit a revised version of the manuscript that addresses the points raised during the review process.

We look forward to receiving your revised manuscript.

Kind regards,

Antoine Naem, M.D.

Academic Editor

PLOS ONE

Journal Requirements:

2. We note that Figure 2 in your submission contain map images which may be copyrighted. All PLOS content is published under the Creative Commons Attribution License (CC BY 4.0), which means that the manuscript, images, and Supporting Information files will be freely available online, and any third party is permitted to access, download, copy, distribute, and use these materials in any way, even commercially, with proper attribution. For these reasons, we cannot publish previously copyrighted maps or satellite images created using proprietary data, such as Google software (Google Maps, Street View, and Earth). For more information, see our copyright guidelines: http://journals.plos.org/plosone/s/licenses-and-copyright.

(1) You may seek permission from the original copyright holder of Figure 2 to publish the content specifically under the CC BY 4.0 license.  

Reviewers' comments:

Reviewer's Responses to Questions

**Comments to the Author**

1. Is the manuscript technically sound, and do the data support the conclusions?

Reviewer #1: Yes

Reviewer #2: Yes

2. Has the statistical analysis been performed appropriately and rigorously? 

Reviewer #1: Yes

Reviewer #2: Yes

3. Have the authors made all data underlying the findings in their manuscript fully available?

Reviewer #1: Yes

Reviewer #2: No

4. Is the manuscript presented in an intelligible fashion and written in standard English?

Reviewer #1: Yes

Reviewer #2: No

5. Review Comments to the Author

Reviewer #1: This contribution is about demographic analysis of Testicular Torsion (TT). Although the value of this paper is undoubtful, the organization of the paper makes the paper difficult to understand. So, I would like to ask major revision.

1) The introduction includes document which should be placed in the Materials section (line 64-67).

2) This authors repeatedly claims (line 74-75, line 306-307) that this particular epidemiological study reduces testicular removal without clear reasons. If this claim is directly derived from the result of this particular study, discuss it clearly. If not, delete these and similar discussions. Claims must be based on the obtained data. Some other discussions (like line 214-217) should be reconsidered in the same manner.

3) In the discussion, it is difficult to understand which data is derived from this particular research and which data is from others. Consequently, the new findings derived from this particular research is unclear. The authors should separately denote discussions from previous researches and ones from this analysis.

4) Even the limitations of this paper clearly stated, the one important issue is whether the given group of ICD codes is exactly match with TT. As the authors pointed out, the disease names given in claim-based database such as Japanese NDB always include some pseudo-diseases to justify certain claims, and, on the contrary, several true-diseases are not denoted in case there are clear needs to denote it for claims point of view. Additionally, testicular pain and scrotal pain which can be coded part of N508, although the authors claim they cannot be coded. Although this is the basic nature of claim-based database, the authors should clearly and carefully denote these possibilities as the limitations.

5) The Acknowledgement explains the reason why the authors cannot share the data. However, the way of writing is too simple to let the readers understand the reason why Japanese Government ban sharing the data of NDB. The authors should refer previous researches using NDB for research or refer any document publicly shared from MHLW. The authors can even ask MHLW how to explain in Acknowledgements o the academic outcome of NDB-based research.

Reviewer #2: This study contains useful information for clinicians that clarifies the incidence of TT and the rate of testicular removal in Japan, which has not been clarified before. However, in the discussion, there are many redundant discussions that are not directly related to the results of this study. Therefore, the discussion should be reviewed in its entirety. In addition, the conclusions the authors state from the present results are too far-fetched. Other points that should be corrected are described below.

It is not clear what is meant by the different colors in Figure 2. The intent of the shading should be indicated. Also, for people around the world, the occurrence in a small Japanese region may not be of interest. Results compared by region should also be shown if they are meaningful to other countries, but if not, it is better to show them in supplimental. The authors should be aware that this is an international journal.

Figure 4: The number of occurrences in each year should not be shown in a stacked bar chart. If the increase or decrease in each year is not evaluated, it should be shown as mean ± standard deviation.

Showing Table 3 during the discussion should be avoided and should be removed or sent to the Supplemental.

Although there is an age difference in incidence, we did not understand the clear reason for the difference in removal rates. Figures 3 and 5 should be shown together and discussed based on the evidence.

It shows that winter season is more common, but the definitions of summer and winter are unclear. The rationale should be provided. The definition differs from the general definition and is arbitrary.

Minor issue.

Explanation of the vertical axis in Figure 5 is missing.

6. PLOS authors have the option to publish the peer review history of their article (what does this mean?). If published, this will include your full peer review and any attached files.

Reviewer #1: **Yes: **Tomohiro Kuroda

Reviewer #2: No

---

## [Author Response · Author response to Decision Letter 0]

29 Nov 2023

Comments to the Reviewers

I will provide details regarding the changes made in response to the points you pointed out.

Reviewer #1: This contribution is about demographic analysis of Testicular Torsion (TT). Although the value of this paper is undoubtful, the organization of the paper makes the paper difficult to understand. So, I would like to ask major revision.

1) The introduction includes document which should be placed in the Materials section (line 64-67).

→ Thank you for your insightful comment. Following the review of several relevant studies, we have incorporated an introduction with a brief explanation of the National Database (NDB) (line 77-80). Additionally, we have included the more detailed description of the NDB in the Materials and Methods section (line 85-97).

2) This authors repeatedly claims (line 74-75, line 306-307) that this particular epidemiological study reduces testicular removal without clear reasons. If this claim is directly derived from the result of this particular study, discuss it clearly. If not, delete these and similar discussions. Claims must be based on the obtained data. Some other discussions (like line 214-217) should be reconsidered in the same manner.

→ Thank you for your valuable suggestions. Regarding the indicated statements, we have deleted them because they are based solely on personal opinion.

3) In the discussion, it is difficult to understand which data is derived from this particular research and which data is from others. Consequently, the new findings derived from this particular research is unclear. The authors should separately denote discussions from previous researches and ones from this analysis.

→ Thank you for your pertinent comments. In a single comprehensive paragraph within the discussion section, we initially presented the outcomes of our research. Following this presentation, we conducted a comparative analysis with the results obtained in previous studies, which served as the foundation for our subsequent discussion and interpretation. 

Furthermore, to enhance reader comprehension, we reduced the extent of discussion points, thus simplifying the presentation. We have narrowed down the main discussion points on the incidence rate, orchiectomy rate, and seasonality of testicular torsion (including regional aspects).

4) Even the limitations of this paper clearly stated, the one important issue is whether the given group of ICD codes is exactly match with TT. As the authors pointed out, the disease names given in claim-based database such as Japanese NDB always include some pseudo-diseases to justify certain claims, and, on the contrary, several true-diseases are not denoted in case there are clear needs to denote it for claims point of view. Additionally, testicular pain and scrotal pain which can be coded part of N508, although the authors claim they cannot be coded. Although this is the basic nature of claim-based database, the authors should clearly and carefully denote these possibilities as the limitations.

→ Thank you for your insightful comments. We added the code, N508, indicating testicular pain and scrotal pain (line 124). 

We have further updated the Materials and Methods section to emphasize that, given the typical necessity for surgery following the diagnosis of TT, the purpose of extracting IDs that combine both the procedural code and the disease name was to exclude pseudo diagnoses (line 110-117). We have also included a flowchart, within the constraints of what can be publicly disclosed, to illustrate how the collected data were analyzed in our study; we have made alterations to the content of Figure 1. To enhance clarity, we have provided a breakdown of the obtained data in Table 1. 

Furthermore, we added a section illustrating specific examples to acknowledge the limitations of this study (line 302-312).

5) The Acknowledgement explains the reason why the authors cannot share the data. However, the way of writing is too simple to let the readers understand the reason why Japanese Government ban sharing the data of NDB. The authors should refer previous researches using NDB for research or refer any document publicly shared from MHLW. The authors can even ask MHLW how to explain in Acknowledgements o the academic outcome of NDB-based research.

→ Thank you for your important suggestion. Referring to a previous study, we have expanded in the Methods section information about the anonymization of data within the NDB and a detailed description of the data elements provided (line 104-109). We have also included the guidelines regarding access to data in the Acknowledgements section (line 332-335).

Reviewer #2: This study contains useful information for clinicians that clarifies the incidence of TT and the rate of testicular removal in Japan, which has not been clarified before. However, in the discussion, there are many redundant discussions that are not directly related to the results of this study. Therefore, the discussion should be reviewed in its entirety. In addition, the conclusions the authors state from the present results are too far-fetched. Other points that should be corrected are described below.

It is not clear what is meant by the different colors in Figure 2. The intent of the shading should be indicated. Also, for people around the world, the occurrence in a small Japanese region may not be of interest. Results compared by region should also be shown if they are meaningful to other countries, but if not, it is better to show them in supplimental. The authors should be aware that this is an international journal.

Figure 4: The number of occurrences in each year should not be shown in a stacked bar chart. If the increase or decrease in each year is not evaluated, it should be shown as mean ± standard deviation.

Showing Table 3 during the discussion should be avoided and should be removed or sent to the Supplemental.

Although there is an age difference in incidence, we did not understand the clear reason for the difference in removal rates. Figures 3 and 5 should be shown together and discussed based on the evidence.

It shows that winter season is more common, but the definitions of summer and winter are unclear. The rationale should be provided. The definition differs from the general definition and is arbitrary.

Minor issue.

Explanation of the vertical axis in Figure 5 is missing.

→　Thank you for your valuable suggestions. In accordance with your feedback, the following revisions have been made:

1. Figure 2(→ We have deleted.): In consideration of the international status of the journal, we have removed the map from the main text. Furthermore, we have added a detailed description of the characteristics of Japan's regions and their climates in the main text (line 155-158).

2. Figures 3 and 5(→ We have updated these as Figure 2.): As you suggested, we have combined these figures into a single figure. Additionally, we have addressed the missing vertical axis label in Figure 5 and included a description of the age differences of the extraction rate in the main text (line 262-266).

3. Figure 4(→ We have updated this as Figure 3.): We have changed the figure from a stacked bar graph to separate bar graphs for each year, as you pointed out.

4. Table 3: As per your suggestion, we have removed the table and provided a clearer description in the main text (line 241-247).

5. Main Text: In response to your comments, we have made significant changes to the Discussion section. We have narrowed down the main discussion points on the incidence rate, orchiectomy rate, and seasonality of testicular torsion (including regional aspects). We have also restructured the discussion to present the results of our study and prior research, followed by a detailed analysis of their differences. Furthermore, we have noted that the definition of the seasons was already included in the text but added a reference to the Japan Meteorological Agency's website for clarification (line 163-166).

We believe that these revisions have improved the clarity and comprehensiveness of our research.

---

## [Decision Letter · Decision Letter 1]

15 Jan 2024

A nationwide epidemiological study of testicular torsion: analysis of the Japanese National Database

PONE-D-23-27667R1

Dear Dr. Den,

We’re pleased to inform you that your manuscript has been judged scientifically suitable for publication and will be formally accepted for publication once it meets all outstanding technical requirements.

Kind regards,

Antoine Naem, M.D.

Academic Editor

PLOS ONE

Additional Editor Comments (optional):

Reviewers' comments:

Reviewer's Responses to Questions

**Comments to the Author**

1. If the authors have adequately addressed your comments raised in a previous round of review and you feel that this manuscript is now acceptable for publication, you may indicate that here to bypass the “Comments to the Author” section, enter your conflict of interest statement in the “Confidential to Editor” section, and submit your "Accept" recommendation.

Reviewer #1: All comments have been addressed

Reviewer #2: All comments have been addressed

2. Is the manuscript technically sound, and do the data support the conclusions?

Reviewer #1: Yes

Reviewer #2: Yes

3. Has the statistical analysis been performed appropriately and rigorously? 

Reviewer #1: Yes

Reviewer #2: Yes

4. Have the authors made all data underlying the findings in their manuscript fully available?

Reviewer #1: Yes

Reviewer #2: Yes

5. Is the manuscript presented in an intelligible fashion and written in standard English?

Reviewer #1: Yes

Reviewer #2: Yes

6. Review Comments to the Author

Reviewer #1: Thank you for your sincere corrections. Now I believe your contribution becomes clearer. Your careful discussion may help readers about your research outcomes.

Reviewer #2: (No Response)

7. PLOS authors have the option to publish the peer review history of their article (what does this mean?). If published, this will include your full peer review and any attached files.

Reviewer #1: **Yes: **Tomohiro Kuroda

Reviewer #2: No

---

## [Editor Report · Acceptance letter]

26 Feb 2024

PONE-D-23-27667R1 

PLOS ONE

Dear Dr. Den, 

I'm pleased to inform you that your manuscript has been deemed suitable for publication in PLOS ONE. Congratulations! Your manuscript is now being handed over to our production team.

Kind regards, 

on behalf of

Dr. Antoine Naem 

Academic Editor

PLOS ONE